# GaussianBlock: Building Part-Aware Compositional and Editable 3D Scene by Primitives and Gaussians

**Shuyi Jiang**[1], **Qihao Zhao**[1], **Hossein Rahmani**[2], **De Wen Soh**[1], **Jun Liu**[2], **Na Zhao**[1*]
[1] Singapore Univeristy of Technology and Design, [2] Lancaster University
shuyi_jiang@mymail.sutd.edu.sg,
{qihao_zhao, dewen_soh, na_zhao}@sutd.edu.sg,
{h.rahmani, j.liu81}@lancaster.ac.uk

## Abstract

Recently, with the development of Neural Radiance Fields and Gaussian Splatting, 3D reconstruction techniques have achieved remarkably high fidelity. However, the latent representations learned by these methods are highly entangled and lack interpretability. In this paper, we propose a novel part-aware compositional reconstruction method, called GaussianBlock, that enables semantically coherent and disentangled representations, allowing for precise and physical editing akin to building blocks, while simultaneously maintaining high fidelity. Our GaussianBlock introduces a hybrid representation that leverages the advantages of both primitives, known for their flexible actionability and editability, and 3D Gaussians, which excel in reconstruction quality. Specifically, we achieve semantically coherent primitives through a novel attention-guided centering loss derived from 2D semantic priors, complemented by a dynamic splitting and fusion strategy. Furthermore, we utilize 3D Gaussians that hybridize with primitives to refine structural details and enhance fidelity. Additionally, a binding inheritance strategy is employed to strengthen and maintain the connection between the two. Our reconstructed scenes are evidenced to be disentangled, compositional, and compact across diverse benchmarks, enabling seamless, direct and precise editing while maintaining high quality. The code is available at https://github.com/Jiangshuyi0V0/GaussianBlock.

## 1 Introduction

Recently, with the development of Neural Radiance Fields (NeRF) (Barron et al., 2021; Mildenhall et al., 2021) and Gaussian Splatting (Jiang et al., 2023; Kerbl et al., 2023) approaches, multi-view 3D reconstruction techniques have substantially progressed, enabling the recovery of high-quality 3D assets from multi-view 2D images. Despite impressive progress, the underlying representations learned by these methods are highly entangled and lack interpretability. This entanglement nature not only discourages the understanding and analysis of the model, but also hinders precise editing of the reconstructed assets, as it is difficult to accurately locate the required corresponding representations and apply precise operations while ensuring other representations remain unaffected. Although advanced 3D editing methods can serve as post-processing tools, *e.g.*, one of the current state-of-the-art methods, GaussianEditor (Chen et al., 2024), proposes semantic tracing to constrain the modification region while keeping other parts unchanged, it remains challenging to achieve precise control as shown in Fig. 1. In contrast, compositional reconstruction - where disentangled 3D components are reconstructed from 2D image inputs - allows for individual analysis, processing, and modification, making it a particularly appealing approach. This compositional reconstruction is essential for enhancing 3D perception and understanding (Yuan et al., 2023; Ye et al., 2023), as well as for improving robotic interaction and manipulation capabilities (Vezzani et al., 2017).

A number of prior works have attempted to achieve part-aware compositional reconstruction from 2D inputs; however, several common limitations still exist. **1) Textureless or limited fidelity.** Early

---

*Corresponding Author: na_zhao@sutd.edu.sg

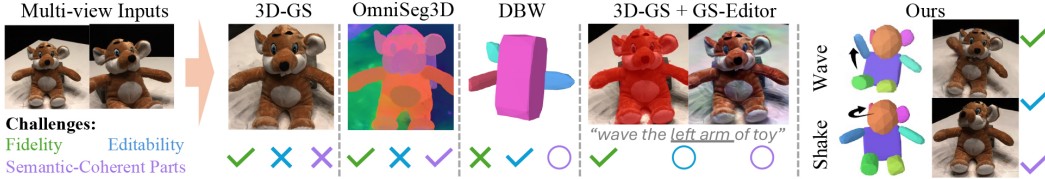

Figure 1: While most related works face at least one of the three common limitations, including fidelity, editability, and semantically coherent part-aware disentanglement. Our method addresses all three limitations simultaneously. The underline text and masks are semtanic tracing keywords and results. Best viewed with color and marks.

classical works focus solely on extracting textureless part-aware shapes from images (Niu et al., 2018; Wu et al., 2020; Paschalidou et al., 2020; Michalkiewicz et al., 2021; Paschalidou et al., 2021; Petrov et al., 2023; Alaniz et al., 2023; Liu et al., 2024). Although some recent works attempt to jointly learn appearance through textured UV-maps (Monnier et al., 2023) or neural features (Yu et al., 2024; Tertikas et al., 2023), the fidelity of the reconstructed scenes still lags significantly behind current reconstruction methods utilizing Gaussian Splatting. **2) Semantic incoherence.** One line of compositional reconstruction works (Monnier et al., 2023; Alaniz et al., 2023; Yu et al., 2024) opts to use superquadrics Paschalidou et al. (2019); Liu et al. (2022), a specific type of geometric primitives to build a 3D scene due to their high interpretability and flexible actionability, which can be directly adapted into industrial software for intuitive physical editing. However, these methods typically focus merely on fitting "geometrically-suited" primitives into 3D scenes, while neglecting the semantic relationships and coherence between compositional parts. This often results in undesirable semantic overlap and unreasonable segmentation among reconstructed primitives. **3) Lack of controllable editablity.** Another line of methods achieves compositional reconstruction through generalizable image segmentation models like SAM (Kirillov et al., 2023). These methods utilize SAM to segment visual components in images and align the generated 2D masks with implicit representations (*e.g.*, NeRF) (Bai et al., 2023; Ying et al., 2024; Cen et al., 2023b; Tertikas et al., 2023) or advanced explicit representations (*e.g.*, Gaussians) (Ye et al., 2023; Cen et al., 2023a; Dou et al., 2024; Ying et al., 2024). Although these approaches yield satisfactory decomposed results thanks to impressive segmentation techniques, the reconstructed scenes only support "holistic" edits, such as translation and deletion, rather than allowing for *controllable editablit*y. For instance, while it is feasible to uniformly remove or translate Gaussians within the same mask, more complex manipulations - such as waving arms or shaking heads, as shown in Fig. 1 - require each Gaussian to undergo distinct transformations, a capability that these methods do not support.

These challenges prompt us to propose a novel reconstruction pipeline, named as *GaussianBlock*, where the semantically coherent and disentangled representation can be precisely and physically edited like building blocks, while simultaneously maintaining competitive high fidelity. Specifically, leveraging the flexible actionability of superquadric primitives and the capacity of 3D Gaussians in reconstructing high-quality scenes, we propose a hybrid representation that integrates superquadric primitives and Gaussians, where superquadric serve as coarse but disentangled building blocks, while Gaussians act as "skin" to refine the structure and ensure high fidelity.

Our GaussianBlock reconstructs semantically coherent primitives by introducing a novel Attention-guided Centering (AC) loss to regularize the generated superquadrics. Specifically, by using 2D differentiable prompts derived from superquadrics through dual-rasterization, the AC loss applies a clustering algorithm to supervise the grouping of queries with the same semantics, encouraging the superquadrics to disentangle accordingly. Additionally, our GaussianBlock enhances the compactness and semantic coherence of superquadrics through the introduction of dynamic fusion and splitting strategies. Specifically, a single superquadric that encompasses various distinguishable semantics is considered for splitting, while multiple superquadrics that share the same semantics or exhibit significant overlap are fused.

The hybrid representation in our GaussianBlock is established by attaching and connecting Gaussians to the superquadrics. Specifically, inspired by GaussianAvater (Qian et al., 2024), we initialize 3D Gaussians at the triangles of each reconstructed superquadric and perform primitive-based localized transformations during rasterization to reinforce their connection. Additionally, a binding

inheritance strategy is employed to maintain this connection, assigning each Gaussian new identity parameters that indicate the primitive and triangle to which it belongs. During optimization and density control, our GaussianBlock utilizes a regularization loss to encourage the Gaussians to remain closely connected to their corresponding primitives while ensuring that these identity parameters are inherited throughout the process. Based on these designs, as primitives are manipulated, each 3D Gaussian can be deleted, translated, and rotated according to its associated binding vertex.

Thanks to these innovative designs, our method demonstrates state-of-the-art part-level decomposition and controllable, precise editability, with competitive fidelity across various benchmarks, including DTU (Jensen et al., 2014), Nerfstudio (Tancik et al., 2023), BlendedMVS (Yao et al., 2020), Mip-360-Garden (Barron et al., 2021) and Tank&Temple-Truck (Knapitsch et al., 2017).

## 2 RELATED WORK

**Compositional Reconstruction.** Compositional reconstruction refers to reconstructing disentangled 3D components from 2D image inputs, which can be individually analyzed, processed, and modified. Early classical works often focus on abstracting textureless shape parts from images, mostly through hierarchical unsupervised learning (Paschalidou et al., 2020), deep neural network (Niu et al., 2018; Paschalidou et al., 2021; Michalkiewicz et al., 2021; Petrov et al., 2023), generative models (Wu et al., 2020), contrastive learning (Liu et al., 2024) or primitives (Alaniz et al., 2023). Recently, there have been two major lines of work in compositional reconstruction: one is primitive-based, and the other is segment-based with advanced representation.

In general, primitive-based methods typically fit "geometry-suited" primitives into the scene. For instance, Monnier *et al.* (Monnier et al., 2023) optimizes the superquadrics and corresponding texture map concurrently, while Alaniz *et al.* (Alaniz et al., 2023) iteratively feeds new superquadrics to an area with high reconstruction error. Recently, DPA-Net (Yu et al., 2024) fuses convex quadrics assembly into NeRF framework to generate an occupancy field. However, existing primitive-based works neglect the semantic relationships and coherence between compositional parts, leading to undesirable semantic overlap and unreasonable segmentation among the reconstructed primitives.

With the advent of large generalizable image segmentation models like SAM (Kirillov et al., 2023), a new line of segmentation-based methods that integrate with advanced 3D scene representations such as NeRF and Gaussian Splatting has gained popularity. PartNeRF (Tertikas et al., 2023) learns independent NeRF for each pre-segmented part. CompoNeRF (Bai et al., 2023) segments complex text prompts and positions sub-texts within different bounding boxes, followed by the learning of a distinct NeRF for each box. In contrast, SA3D (Cen et al., 2023b) utilizes a 3D NeRF as input instead of 2D images. It projects 2D pre-segmented masks onto 3D mask grids via density-guided inverse rendering, thereby achieving a certain level of 3D disentanglement. GaussianGrouping (Ye et al., 2023) aligns rendered Gaussians with their corresponding pre-segmented 2D masks and groups Gaussians with similar meanings to achieve disentanglement. Similarly, SegAnyGaussians Cen et al. (2023a) achieves Gaussian decomposition and segmentation by enforcing the correspondence between Gaussian features and 2D pre-segmented masks. Omniseg3d (Ying et al., 2024) addresses the labor intensity required to ensure that 2D masks are multi-view and fine-grained consistent by introducing 3D hierarchical contrastive learning. Despite the impressive segmentation and decomposition results achieved by the above methods, which rely on pre-segmented masks and advanced 3D representations, the reconstructed 3D assets still lack flexible editability and actionability.

**3D Editing.** Precise editing is a challenging task, especially when dealing with the entangled 3D underlying representations of reconstructed scenes. Recently, numerous advanced 3D editing methods have been proposed, typically taking 3D scenes as input. These methods can function as post-processing techniques to achieve visual editing effects relevant to our primary focus, *i.e.*, compositional reconstruction. Therefore, we discuss several of these works here. Focaldreamer (Li et al., 2024) takes 3D meshes as input and introduces geometry union and focal loss to achieve text-prompted precise editing. SKED (Mikaeili et al., 2023) requires 3D NeRF, sketches, and text prompt, where sketches guide the regions of the NeRF that should adhere to the specified modifications. LENeRF (Hyung et al., 2023) also employs NeRF as input and incorporates several add-on modules, such as the latent residual mapper and attention field generation, to execute text prompts that contain the desired local edits. GaussianEditor (Chen et al., 2024), a leading method in the field, takes well-trained Gaussian scenes and text prompts as inputs, and enhances precise editing through Gaussian semantic tracing, which traces the editing target throughout the training process. Unlike

these methods, our approach enables intuitive "*drag-mode*" physical and precise editing, similar to building blocks, by reconstructing actionable and semantic compositional representations.

## 3 METHOD

Given a set of multi-view images $\mathbf{I}$ and their corresponding silhouettes $\mathbf{I^s}$, our goal is to reconstruct the compositional 3D scene using semantic coherent compositional primitives combined with Gaussians. The overall pipeline is illustrated in Fig. 2.

In the first stage, we adopt parameterized superquadrics primitives $\Theta_K$, for their remarkable expressiveness, which could be achieved with a relatively small number of continuous parameters (Barr, 1981; Paschalidou et al., 2019). In order to query from 2D pretrained segmentaion model and obtain semantic prior, we can derive the differential point $P_k$ and box prompts $B_k$ from parameterized $\Theta_K$ through soft dual-rasterization, as derived in Sec. 3.1. Subsequently, with the reference image $\mathbf{I}$ and prompts, we can extract the corresponding attention maps $A_k$ for each $P_k$ through pretrained prompt encoder $PE$, image encoder $E$ and deconder $D$, and then apply clustering to $A_k$ to derive a novel AC loss, as described in Sec. 3.2, where outlier vertices of the superquadrics are encouraged to move toward the centroid (indicated as star in Fig. 2). Additionally, splitting and fusion strategy, explained in Sec. 3.3, is proposed to further strengthen the compactness and disentanglement, when applying the AC loss and reconstruction loss between reconstructed image $\mathbf{I}^r$ and silhouette $\mathbf{I}^s$ during the supervised optimization. Through the first stage, semantic coherent part-aware superquadrics are obtained. In the second stage, to leverage the capacity of 3D Gaussians, Gaussians are initialized and bound to superquadric triangles with localized parameterization, and will be globally mapped to world space during rasterization. Additionally, a regularization term $L_{pos}$ will be applied together with $L_{rgb}$, as introduced in Sec. 3.4, to enhance the connection between them.

### 3.1 DIFFERENTIAL RENDERING

**Superquadric Parameterization.** Following the classic work (Barr, 1981), superquadrics can be parameterized as $\Theta_K = \{\alpha_K, r_K, t_K, s_K, \epsilon_K\}$, where $K$, $\alpha \in \mathbb{R}$, $r \in \mathbb{R}^6$, $t \in \mathbb{R}^3$, $s = \{s_1, s_2, s_3\} \in \mathbb{R}^3$, and $\epsilon = \{\epsilon_1, \epsilon_2\} \in \mathbb{R}^2$ denote the number of superquadrics, transparency, 6D rotation vector, displacement vector, scaling factor, and continuous shape vector, respectively. Specifically, superquadrics can be formulated as the following explicit surface function (Barr, 1981):

$$\Phi(x; \Theta_k) = \begin{bmatrix} x_1 \\ x_2 \\ x_3 \end{bmatrix}^T \begin{bmatrix} s_1 \cos^{\epsilon_1} \eta \cos^{\epsilon_2} \omega \\ s_2 \sin^{\epsilon_1} \eta \\ s_3 \cos^{\epsilon_1} \eta \sin^{\epsilon_2} \omega \end{bmatrix}, \tag{1}$$

$$\text{where } x = \mathbf{rot}(r_k)x + t_k. \tag{2}$$

Here $\eta \in [-\frac{\pi}{2}, \frac{\pi}{2}]$ and $\omega \in [-\pi, \pi]$ are the spherical coordinates of the initial unit icosphere vertices, and $\mathbf{rot}(\cdot)$ is a rigid transformation (Zhou et al., 2019) mapping the 6-D rotation vector to a rotation matrix.

**Soft Dual Rasterization.** In order to query the 2D pretrained segmentation model and obtain semantic priors, we propose soft *dual* rasterization, which means we rasterize both rendered images $\mathbf{I}^r$ and 2D prompts $P_k$ derived from the superquadrics simultaneously.

To render the superqudaric, we adopt the occupancy function of (Chen et al., 2019) and attach the superquadric surface $\Phi(x; \Theta_k)$ with transparency information $\alpha_k$ (Monnier et al., 2023), which can be summarized as:

$$O_j(u) = \alpha_{k_j} \exp\left(\min\left(\frac{\Delta_j(u)}{\sigma}, 0\right)\right). \tag{3}$$

Given the pixel $u$ and projected face $j$, $\Delta_j(u) < 0$ indicates that pixel $u$ is outside face $j$, and vice versa; $\sigma$ serves as a scalar modeling the extent of the soft mask. Based on the occupancy function, we can associate $L$ faces per pixel and obtain the rendered image $\mathbf{I}^r$ through the classic alpha compositing (Porter & Duff, 1984):

$$\mathbf{I}^r(O_{1:L}) = \sum_{l=1}^{L} \left(\prod_{p<l} (1 - O_p)\right) \odot O_l. \tag{4}$$

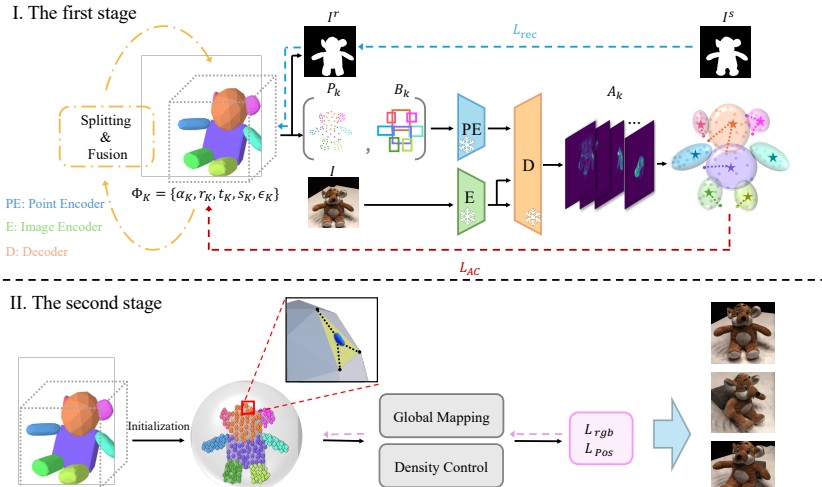

Figure 2: **The framework of our pipeline**. In the first stage, the superquadrics blocks are optimized guided by reconstruction loss $\mathcal{L}_{rec}$ and attention-guided centering loss $\mathcal{L}_{AC}$. With the point $P_k$ and bounding box $B_k$ prompt obtained from soft dual rasterization, the attention maps $A_k$ from the last layer of pretrained decoder D are clustered, where outliers being encouraged to move towards the centroid. Meanwhile, superquadric splitting and fusion strategy is proposed to further enhance the semantic coherence and compactness during optimization. In the second stage, Gaussians are bound to the triangles of superquadrics using localized parameterization and inheritance strategy duing optimization and densification with global mapping transform.

Additionally, to obtain the reference points prompts as well, all superquadric vertices $V$ can be projected to screen coordinate using the following transformation:

$$\Psi(V) = \bigcup_{k=1}^{K} \mathbb{M}_{Ns} \left( \frac{\mathbb{M}_{Pj} \cdot \mathbb{M}_{Vi} \cdot \begin{bmatrix} V \\ 1 \end{bmatrix}}{\tilde{\omega}} \right), \tag{5}$$

where $\mathbb{M}_{Pj}$ and $\mathbb{M}_{Vi}$ are projection and view matrices derived from camera's position and orientation (Ravi et al., 2020), $\tilde{\omega}$ is the resulting component from projection transformation, and $\mathbb{M}_{Ns}$ accounts for the conversion from normalized device coordinates (NDC) space to screen coordinates. Unfortunately, such an aggressive transformation incorporates undesirable vertices, whose positions are occluded due to superquadric overlapping from various perspectives. Consequently, these invisible vertices do not "contribute to" the final rendered image, causing them too ambiguous to serve effectively as point prompts. To obtain valid point prompts, we randomly assign each superquadric a unique representation color $\{C_k | k \in [1, K]\}$, and write the vertices rasterization equation as:

$$P_k = \left\{ \Psi(V_k) \mid V_k \in \left[ \sum_{l=1}^{L} \mathbf{I}^r \left( \Psi(V_k) \right) \odot C_l \bigcap C_k \right] \right\}, \tag{6}$$

where $V_k$ denotes the $k$-th superquadric's vertices and $\{C_l | l \in L\}$ represents the color attribute associated with the $l$-th intersecting face. Vertices are considered as point prompts only when they are meaningful, *i.e.*, when the representative color of a vertex is visible in the final outcome.

## 3.2 ATTENTION-GUIDED CENTERING

Aiming for semantic coherence, we expect the superquadrics to be disentangled in alignment with the semantic parts of the image **I**. In this work, we utilize SAM (Kirillov et al., 2023) to provide the semantic prior, given its advanced performance and generalizability. Based on this semantic prior, we propose a novel Attention-guided Centering (AC) loss to ensure alignment. The overall procedure for calculating the AC loss is summarized in Algorithm 1.

Firstly, to trigger the pre-trained SAM model, bounding boxes and point prompts are typically required along with the input images for localization and indication. To be specific, for each su-

---

**Algorithm 1** Attention-guided Centering Loss

---

**Input:** a set of bounding boxes $B_k$ and 2D point prompts $P_k$, image embedding $h^{(\mathbf{I})}$
**Output:** the AC loss $\ell_k$ for superquadric $\theta_k$
1: $\mathcal{A}_k = \varnothing$
2: **for** $p_{k,m} \in P_k$ **do**
3:      $h_{k,m}^{(p)} \leftarrow f^{(p)}([p_{k,m}, B_k])$              ▷ encode point and box prompt
4:      $A_{k,m} = \text{CrossAttention}(h^{(\mathbf{I})}, h_{k,m}^{(p)})$          ▷ get cross attention map
5:      $\mathcal{A}_k \leftarrow \mathcal{A}_k \cup \{A_{k,m}\}$
6: **end for**
7: $\mathcal{S}_k \leftarrow \text{HDBSCAN}(\mathcal{A}_k)$                        ▷ cluster $\mathcal{A}_k$
8: $S_k^* = \arg\max_{S_{k,n} \in \mathcal{S}_k} |S_{k,n}|$ and $\mathcal{M}_k^* = \{m | A_{k,m} \in S_k^*\}$    ▷ find the major cluster $S_k^* \in \mathcal{S}^{(k)}$
9: $\bar{A}_k^* \leftarrow \frac{1}{M} \sum_{m \in \mathcal{M}_k^*} (\Pr[\tilde{n}_{k,m}|A_{k,m}] \cdot A_{k,m})$      ▷ weighted average among $S_k^*$
10: $m_k^* = \arg\min_{m \in \mathcal{M}_k^*} \mathbf{d}(A_{k,m}, \bar{A}_k^*)$          ▷ localize the centroid of $S_k^*$
11: $P_k^\dagger = \complement_{P_k} \{p_{k,m}|A_{k,m} \in S_k^*\}$            ▷ define outliers
12: $\ell_k \leftarrow \sum_{p_{k,i} \in P_k^\dagger} \mathbf{d}(p_{k,i}, p_{k,m_k^*})$         ▷ calculate the AC loss

---

perquadric $\theta_k$, its rendering can be acquired through Eq. 5 as $\mathbf{I}_k$, and corresponding box prompt $B_k$ can be inferred from the top-left and top-right vertices of $\mathbf{I}_k$. Thus, the prompt set of $\theta_k$ can be denoted as $\{[p_{k,m}, B_k] | p_{k,m} \in P_k, k \in K\}$, and the embedding of each prompt $h_{k,m}^{(p)}$ is attained through the pre-trained prompt encoder $f^p$ in SAM. Similarly, the embeddings of the input image denoted as $h^{(\mathbf{I})}$ are obtained through the image encoder $\mathbf{E}$.

Subsequently, we retrieve the attention map $A_{k,m}$ prompted by $h_{k,m}^{(p)}$ from the last cross attention layer of pretrained decoder $\mathbf{D}$. The overall attention map set of $\theta_k$, denoted as $\mathcal{A}_k$, is assembled from all the prompts within $\{[p_{k,m}, B_k]\}$. Since we aim for each superquadric to be disentangled and independent, we expect only one semantic cluster to be identified in each $\theta_k$. Hence, we leverage HDBSCAN (Campello et al., 2015) algorithm to cluster the sets of attention map $\mathcal{A}_k$. Let $\mathcal{S}_k = \{S_{k,n} | n = 1, \ldots, N\}$ denote the cluster set associated with $\theta_k$, and let $\Pr[n|A_{k,m}]$ denote the probability of assigning $A_{k,m}$ to a specific cluster $S_{k,n}$. The corresponding cluster label for $A_{k,m}$, denoted as $\tilde{n}_{k,m}$, is determined by selecting the cluster with the highest probability: $\tilde{n}_{k,m} = \arg\max_n \Pr[n|A_{k,m}]$. The major cluster, denoted as $S_k^*$, is defined as the largest cluster among $\mathcal{S}_k$, where $S_k^* = \arg\max_{S_{k,n}} |S_{k,n}|$, and its counterpart is denoted as the $\complement_{S_k} S_k^*$. Likewise, the centroid $p_{k,m_k^*}$ represents the coordinate whose $A_{k,m}$ is nearest to the weighted average attention map $\bar{A}_k^*$ among $S_k^*$, while the outliers $P_k^\dagger$ are those belonging to $\complement_{S_k} S_k^*$.

Finally, to promote disentanglement, outliers are motivated to move towards the centroid, which can be expressed mathematically as:

$$\mathcal{L}_{\text{AC}}\left(h, \{B_k\}_{k=1}^K, \{P_k\}_{k=1}^K\right) = \sum_{k=1}^K \ell_k; \quad \ell_k = \sum_{p_{k,i} \in P_k^\dagger} \mathbf{d}(p_{k,i}, p_{k,m_k^*}), \quad (7)$$

where $\mathbf{d}$ stands for the L2 Euclidean distance function.

### 3.3 DYNAMIC FUSION AND SPLITTING

To further enhance semantic coherence and compactness while preventing local minima, this section introduces dynamic superquadric splitting and fusion, taking semantic and compact constraints into account.

**Splitting.** We periodically check whether a single primitive contains multiple distinguishable semantics, and for such primitives, we apply a splitting operation. Moreover, due to the centering action motivated by $L_{AC}$, superquadrics tend to cover toward containing single disentangled semantic, occasionally resulting in local minima. Inspired by (Kerbl et al., 2023), we propose a dynamic splitting algorithm tailored for superquadrics, informed by the attention-based centroids.

Based on the cluster set $\mathcal{S}_k$, centroids $m_{k,j}$ can be localized following Algorithm 1, and the distances between each pair of centroids can be denoted as $\{D_{i,j}^k | i, j \in [1, N]\}$. If the furthest distance

$D_{i^*,j^*}^k$ among $D_{i,j}^k$ exceeds the threshold $\tau_s$, which is set according to diagonal of $B_k$ as $\beta \check{B}_k$, then this superquadric $\theta_k$ shall be split, anchored at 3D vertices $v_{i^*}^k$ and $v_{j^*}^k$. In other words, two new superqudrics $\{\theta_z | z \in [K+1, K+2]\}$ will be initialized to replace $\theta_k$, whose $r_z$, $\alpha_z$, and $\epsilon_z$ are inherited from $\theta_k$, and $s_z = 0.4 s_k$. Notably, to anchor the new superquadrics towards $\{v_x^k | x \in [i^*, j^*]\}$, $t_z$ is initialized as Eq. 8 such that their centers are positioned at $v_x^k$, where $\bar{V}_k$ is the average position of $\theta_k$:

$$t_z = v_x^k - \left( \left( \bar{V}_k \circ s_z \right) \odot \mathbf{rot}(r_z) \right). \tag{8}$$

Finally, the opacity $\alpha_k$ is set to 0, indicating complete transparency.

**Fusion.** In addition to splitting, we enable dynamic superquadric fusion to enhance the compactness. Assuming that superquadric $\theta_e$ and $\theta_g$ share the same semantic information, it is supposed that they should be fused together. Particularly, when only one cluster appears in the attention map union $\mathcal{A}_e \cup \mathcal{A}_g$ across all camera views, a new $\theta_{K+1}$ will be initialized to represent the fused one.

Similar to the splitting process, $r_{K+1}$, $t_{K+1}$, $\epsilon_{K+1}$, $\alpha_{K+1}$, and $s_{K+1}$ are determined as the mean corresponding values of $\theta_e$ and $\theta_g$. Since the initial fused outcomes are quite coarse, this leads to a sudden increase in reconstruction error and associated gradients, which heightens the instability of the optimization and impacts convergence. Considering this, before proceeding with global optimization, we temporarily freeze the gradients of $\Theta_K$ and independently optimize $\theta_{K+1}$ for $\xi$ steps to achieve a more precise fused geometry. Likewise, $\alpha_e$ and $\alpha_g$ are ultimately set to zero.

## 3.4 3D GAUSSIANS BINDING

To leverage the advantages of both the actionability of superquadric and the ability of 3D Gaussians to reproduce high-quality scenes, this section establishes, enhances, and maintains connections between them.

**Gaussian Binding and Inheritance.** Our compositional superquadrics primarily focus on objects within the silhouette. To represent the remaining space, following the approach outlined in (Monnier et al., 2023), we incorporate a centered icosphere primitive to model the background dome and a plane primitive to represent the ground outside the reconstructed superquadrics, as illustrated in Fig. 2. Subsequently, inspired by GaussianAvaters (Qian et al., 2024), we initialize Gaussians $\mathcal{G}$ at each triangle face centers $\{T_k^m | k \in K, m \in M\}$ of the superquadrics, defined as $\mathcal{G}_{n \in N} = \{\mu, q, s, \alpha, c, id_k, id_t\}$. Here, $N$, $M$, and $K$ represent the number of 3D Gaussians in the scene, the primitives, and the faces within the primitives, respectively. The parameters $\mu$, $q$, $s$, $\alpha$, and $c$ denote the Gaussian's position, quaternion, scaling factor, opacity, and spherical harmonic (SH) coefficient, respectively. Additional parameters $id_k$ and $id_t$ are introduced to specify the corresponding superquadric and face index to which each Gaussian is bound.

During rasterization, localized transformations are applied, where each Gaussian is transformed according to the parameters of its bound triangle and superquadrics before rendering. Specifically, we map these properties into the global space by:

$$\mu' = \mu + \mu_t, \quad r' = r \cdot \mathrm{Q}(r_t), \tag{9}$$

where $\mu_t$ is the world position of each triangle center; $r_t$ is concatenated by the direction vector of one of the edges, the normal vector of the triangle, and their cross product, describing the orientation of the triangle in global space as (Qian et al., 2024); $\mathrm{Q}(\cdot)$ transform the rotation vector $r_t$ to quaternions following the (Zhou et al., 2019).

Meanwhile, following the density control strategy in Gaussian Splatting (Kerbl et al., 2023), Gaussians shall also be split or cloned within the primitive-based local space, while inheriting their identity parameters, $id_k$ and $id_t$, to the new Gaussians. Based on the aforementioned designs, as primitives undergo manipulation, each 3D Gaussian can be deleted, translated, and rotated according to its corresponding binding triangle.

**Gaussian Positional Regularization.** Furthermore, a fundamental assumption behind binding is that the Gaussian should roughly match the underlying primitives. For example, a Gaussian representing the body should not be rigged to a triangle on the cheek, as it may hinder disentanglement and subsequent editing work. Therefore, to enhance the connection between Gaussians and primitives, we implement a simple yet effective regularization loss to encourage a compact relationship between the two throughout the optimization process, as

$$\mathcal{L}_{pos} = \big\| max\big( \|\mu\|_2, \epsilon_{pos} \big) - \epsilon_{pos} \big\|_2. \tag{10}$$

Here $\epsilon_{pos}$, set to 0.5, is a threshold that allows Gaussians a certain degree of freedom in their position, enabling them to better refine the primitive-based 3D structure.

## 3.5 OPTIMIZATION DETAILS

**The first stage.** The overall objective function for superquadric optimization is summarized as:

$$\mathcal{L}_{first} = \mathcal{L}_{rec} + \gamma\mathcal{L}_{AC}, \tag{11}$$

where $\mathcal{L}_{rec} = \mathbf{d}(\mathbf{I}^s, \mathbf{I}^r)$ is the reconstruction loss between rendering outcome $\mathbf{I}^r$ and input silhouette $\mathbf{I}^s$; $\gamma$ is the weight of $\mathcal{L}_{AC}$, which equals to 0 during first 2k warm-up iterations. Additionally, splitting and fusion operations are performed every 1k iterations. Meanwhile, we automatically filter out nearly transparent superquadrics, whose $\alpha$ is less than 0.1 following (Monnier et al., 2023).

**The second stage.** The overall objective function for 3D Gaussians binding and optimization is summarized as:

$$\mathcal{L}_{second} = \mathcal{L}_{rgb} + \mathcal{L}_{pos}, \tag{12}$$
$$where \quad \mathcal{L}_{rgb} = (1-\lambda)\mathcal{L}_1 + \lambda\mathcal{L}_{D-SSIM}, \tag{13}$$

with $\lambda = 0.2$ following (Kerbl et al., 2023).

## 4 EXPERIMENT

The experimental setup is detailed in Sec.7.1 of the Appendix.

### 4.1 MAIN RESULTS

**Primitives Reconstruction.** Experiments are conducted on DTU scenes to compare the reconstruction performance of our learnt primitives with SOTA 3D decomposition methods (Liu et al., 2022; Monnier et al., 2023). Notably, EMS (Liu et al., 2022) is a decomposition method that utilizes the ground truth point cloud during optimization, while DBW (Monnier et al., 2023) is the only published primitive-based reconstruction method from 2D images that considers both decomposition and texture.

Tab. 1 shows the CD comparison results for each scene of the DTU dataset, where our reconstructed superquadric achieves the best performance among most scenes with the smallest Chamfer Distance.

| Method | Input | S24 | S31 | S45 | S59 | S63 | S83 | S105 | $\overline{\text{CD}}\downarrow$ |
|---|---|---|---|---|---|---|---|---|---|
| EMS (Liu et al., 2022) | 3D GT | 6.77 | 5.93 | 6.91 | 3.50 | 4.72 | 7.25 | 6.10 | 5.67 |
| DBW (Monnier et al., 2023) | Image | 4.90 | 3.13 | 3.86 | 4.52 | 5.02 | 4.12 | 6.48 | 4.78 |
| **Ours** | Image | **4.21** | **2.25** | **3.15** | **3.35** | **3.12** | **3.82** | **3.95** | **3.69** |

Table 1: Performance Comparison of Primitives Reconstruction Quality on DTU Scenes. Our primitives achieve the best reconstruction quality in all scenes, despite relying only on 2D images.

**Part-aware Semantic Coherent Decomposition.** Fig. 3 shows the results of our method, OmniSeg3D (Ying et al., 2024) and DBW on the DTU (first 2 rows) and BlendedMVS (last 2 rows) datasets. OmniSeg3D is a current SOTA method for part segmentation, capable of generating 3D masks containing different semantic information (indicated by colors) from 2D inputs.

From the results, we observe that DBW exhibits unnatural decomposition and fails to segment the scene into semantically coherent parts due to the lack of semantic guidance. For example, it incorrectly represents both the feet and hands of the Smurf using the same superquadric. Additionally, without a self-adaptive splitting strategy like ours, it tends to fall into local minima, as seen with the toy, where one ear is not represented. For the same reason, DBW is sensitive to the initial blocks number $K$; in complex scenes like Gundam, it requires up to 50 initial blocks to achieve an acceptable result. Compared to OmniSeg3D, our method achieves finer-grained semantic disentanglement. For example, in the Smurf and Gundam scene, OmniSeg3D tends to assign a single semantic mask to the entire instance, whereas our approach decomposes the scene into finer-grained parts.

**Precise Editablity.** In Fig. 4, we demonstrate the precise part-level editability of our method. For each data, we provide visualized results in three columns: part-aware decomposition, primitives editing (through MeshLab), and scene editing.

**Reconstruction Fidelity.** To evaluate the fidelity of our GaussianBlock, we compare it with one of the current state-of-the-art reconstruction methods, Gaussian Splatting (Kerbl et al., 2023), and

| Dataset | Se. | Edit | DTU | | | BlendedMVS | | | Truck | | | Garden | | |
|---------|-----|------|-----|-----|-----|-----|-----|-----|-----|-----|-----|-----|-----|-----|
| Method | | | SSIM↑ | PSNR↑ | LPIPS↓ | SSIM↑ | PSNR↑ | LPIPS↓ | SSIM↑ | PSNR↑ | LPIPS↓ | SSIM↑ | PSNR↑ | LPIPS↓ |
| 3DGS | ✗ | ✗ | 98.3 | 35.7 | 8.8 | 94.4 | 33.8 | 9.2 | 84.7 | 25.0 | 17.3 | 86.2 | 27.0 | 10.1 |
| OmniSeg | ✓ | ✗ | 97.7 | 33.6 | 9.4 | 93.2 | 29.4 | 10.4 | 82.6 | 23.4 | 18.0 | 85.8 | 25.9 | 13.0 |
| DBW | ✗ | ✓ | 71.3 | 19.6 | 26.5 | 49.2 | 16.4 | 41.0 | - | - | - | - | - | - |
| **Ours** | ✓ | ✓ | 94.3 | 30.7 | 10.3 | 90.8 | 28.8 | 11.2 | 81.3 | 22.0 | 20.7 | 84.8 | 23.1 | 16.9 |

Table 2: Quantitative Comparison on reconstruction fidelity.

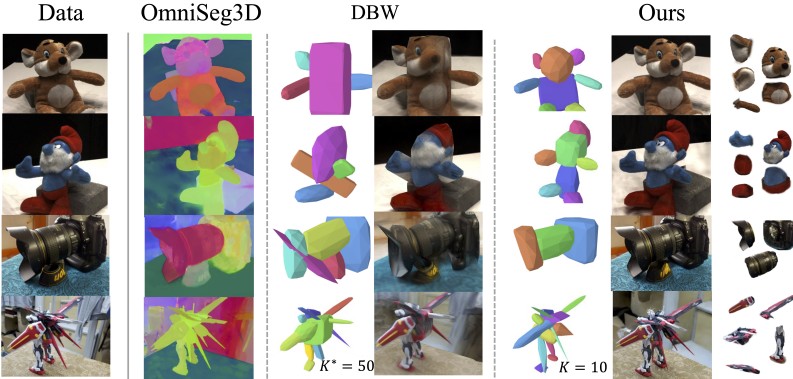

Figure 3: Qualitative results. Our method demonstrates fine-grained, semantically coherent part-aware decomposition.

the Gaussian-based OmniSeg3D, primtive-based DBW in the Tab. 2. Although our method does not achieve the same or better fidelity as purely Gaussian-based approaches, it has made significant improvements over the SOTA primitive-based DBW method, and has produced competitive results compared to Gaussian-based approaches. However, the primary contribution of our method is not to propose a fidelity-oriented reconstruction algorithm. Instead, our method enables semantically coherent part-level decomposition and controllable actionability, with high but not the best fidelity.

### 4.2 ABLATION STUDY

**Attention-guided Centering Loss.** In Tab. 3, we assess the importance of each key component of our model by sequentially removing them and evaluating the average performance across the DTU dataset. Additionally, the total number (#K) of effective superquadrics is presented for reference. The #K can reflect the compactness of the decomposition performance; a common goal is to achieve the highest compactness (the smallest #K) while maintaining the primitives performance.

Overall, $\mathcal{L}_{AC}$ and splitting strategy consistently improve the primitives quality through semantic guidance, while the fusion strategy successfully enhances the compactness decomposition with a minor quality trade-off. An intuitive visual comparison is shown in Fig. 5 (a). While the model without splitting aggressively uses a single superquadric to represent both the roof and the building, the model without fusion redundantly uses multiple superquadrics to represent the same part. This lack of compactness makes subsequent skeleton-based editing less intuitive and meaningful.

**Loss Weight.** In Tab. 4 and Fig. 5 (b), we assess the impact of the weight of $\mathcal{L}_{AC}$, denoted as $\gamma$, on the DTU dataset. It is evident that with a lower $\gamma$, $\mathcal{L}_{AC}$ is too weak to ensure that the superquadrics

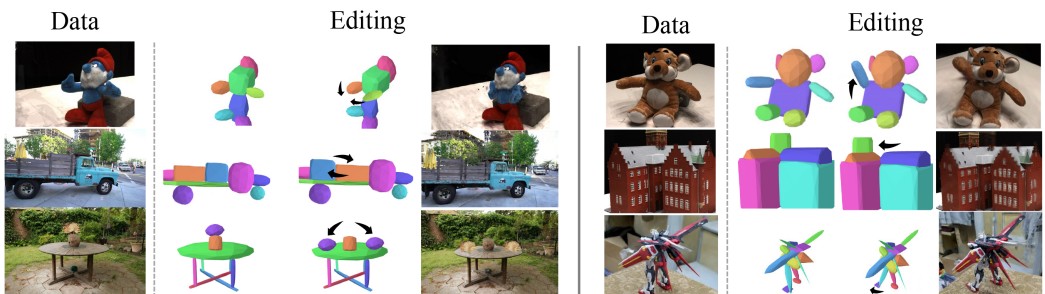

Figure 4: Editing Results. Our reconstructed scenes can be edited seamlessly and precisely.

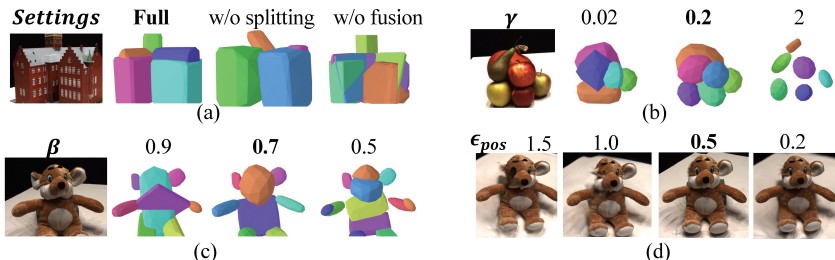

Figure 5: Ablation studies on (a): Experiment settings, (b) AC-Loss weight $\gamma$, (c): Splitting threshold $\beta$; (d): Regularization hyper-parameter $\epsilon_{pos}$

can disentangle the semantic structural information in the input images. Conversely, when $\gamma$ is too high, the superquadrics tend to be small to 'hack' $\mathcal{L}_{AC}$, failing to meet reconstruction expectations. Thus, in this work, $\gamma$ is set to 0.2 by default.

**Splitting Threshold.** Additionally, given that the splitting threshold is $\beta\check{B}_k$, where $\check{B}_k$ is the corresponding diagonal of superquadric $\theta_k$ reconstruction and is dynamically updated during the optimization process, the impact of the weight $\beta$ needs to be evaluated. As shown in Fig. 5 (c), with a higher $\beta$, the splitting mechanism is less likely to be triggered. This results in the toy's body and ears in (c) not being effectively separated into individual superquadrics. Conversely, with a lower threshold, the scene is divided more fine-grained, as seen with the arms and paws. In conclusion, $\beta$ is used to control the granularity of the deompositin and should be selected based on the desired level of granularity for subsequent editing operations, which is set to 0.7 by default.

| Method | #K | CD↓ | SSIM↑ | PSNR↑ | LPIPS↓ |
|---|---|---|---|---|---|
| Full model | 5.8 | 3.69 | 94.3 | 30.7 | 10.3 |
| w/o $\mathcal{L}_{AC}$ | 5.4 | 4.75 | 84.1 | 30.3 | 11.4 |
| w/o Splitting | 4.3 | 4.24 | 88.6 | 25.7 | 16.5 |
| w/o Fusion | 9.2 | 3.46 | 92.5 | 31.0 | 10.1 |

Table 3: Ablation studies on key components.

| | Method | #K | CD↓ | SSIM↑ | PSNR↑ | LPIPS↓ |
|---|---|---|---|---|---|---|
| | 0.02 | 6.2 | 4.12 | 84.5 | 27.2 | 12.4 |
| $\gamma$ | 0.2 (default) | 5.8 | 3.69 | 94.3 | 30.7 | 10.3 |
| | 2 | 4.9 | 6.97 | 80.4 | 20.8 | 13.7 |
| | 0.5 | 9.1 | 3.56 | 90.4 | 31.7 | 10.1 |
| $\beta$ | 0.7 (default) | 5.8 | 3.69 | 94.3 | 30.7 | 10.3 |
| | 0.9 | 4.4 | 4.85 | 84.3 | 32.9 | 14.8 |

Table 4: Ablation studies on $\beta$s and $\gamma$s.

**Positional Regularization Threshold.** We also examined the impact of different thresholds $\epsilon_p os$ in regularization in Fig. 5 (d). Given $\epsilon = 1.5, 1.0, 0.5$, and $0.2$, the PSNR of the reconstructed scenes are 35.4, 33.7, 32.8, 27.3, respectively. However, during the "delete left ear" editing operation, we observe that higher $\epsilon$ tends to perform better. This is because, with a larger threshold, the connection between the primitives and Gaussians becomes weaker, leading to greater intertwinement between Gaussians. For instance, the Gaussian points on the toy's ear semantic component may become entangled with those in the head region, causing the deletion of the ear to affect some undesired parts. On the other hand, when $\epsilon$ is too low, the fidelity suffers, as stronger regularization constraints prevent the Gaussians from fully refining the primitives' structures.

## 5 CONCLUSION, LIMITATIONS AND FUTURE WORK

In this paper, we propose a novel part-aware compositional reconstruction method, called Gaussian-Block, which enables semantically coherent and disentangled representations, allowing for precise and physical editing akin to building blocks, while simultaneously maintaining high fidelity. However, our work has certain limitations. Currently, we can only perform decomposition for object-centric entities based on their silhouette and are not yet able to handle backgrounds effectively. This is due to the fact that backgrounds typically have more complex and detailed structures with richer semantic information. In the future, beyond mid-level superquadrics, we plan to introduce more diverse types of primitives, such as cuboids and curved surfaces, in combination with Gaussians.

## 6 ACKNOWLEDGMENT

This research is supported by the Ministry of Education, Singapore, under its MOE Academic Research Fund Tier 1 - SMU-SUTD Internal Research Grant (SMU-SUTD 2023_02_09), it is also supported by the Agency for Science, Technology and Research (A*STAR) under its MTC Programmatic Funds (Grant No. M23L7b0021).

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

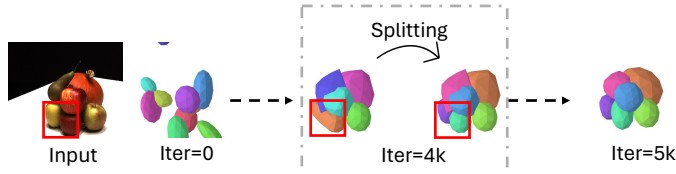

Figure 6: Superquadric Splitting Visualization.

# 7 APPENDIX

## 7.1 EXPERIMENTAL SETUP

**Datasets.** To explore the reconstruction quality as well as illustrate the compositional ability, several benchmark datasets (Jensen et al., 2014; Tancik et al., 2023; Yao et al., 2020; Knapitsch et al., 2017; Barron et al., 2022) are employed in evaluation. The DTU dataset (Jensen et al., 2014) and BlendedMVS (Yao et al., 2020) are MVS benchmarks, encompasses diverse calibrated indoor and outdoor scenes with annotated 3D ground truth. Nerfstudio (Tancik et al., 2023), Mip-360 (Barron et al., 2021) and Tank&Temple (Knapitsch et al., 2017) offers a diverse collection of multi-view real image data that spans dramatically different points in the image space.

**Evaluation Metrics.** Since the main challenges we focus on include part-level internal semantic coherence, editability, and fidelity, we will primarily evaluate the performance in these aspects.

Firstly, due to the limited availability of part-annotated multi-view real datasets, we use Chamfer Distance (CD) (Monnier et al., 2023; Jensen et al., 2014) as an auxiliary metric to quantitatively evaluate the quality of the reconstructed primitives, answering the question of how well the primitives fit the scenes. CD computes the Chamfer distance between the ground-truth points and points sampled from the reconstructed primitives on selected DTU scenes, as our baseline DBW (Monnier et al., 2023). Additionally, since semantic coherence and precise editablity are mainly subjective aspects, we will present qualitative results from all five datasets for subjective evaluation. Finally, to prove the competitive fidelity of our reconstructed scene, we will also use common metrics for quantitative evaluation, including SSIM, PSNR and LPIPS.

**Implementation Details.** In the primitive optimization step, the initial $K$ is set to 10, while the input image is resized to $400 \times 300$ as (Monnier et al., 2023). Approximately 50k iterations are performed for each scene. Besides, the splitting threshold $\beta$, the fusion warm up steps $\xi$ and the weight $\gamma$ of the $\mathcal{L}_{AC}$ are set to 0.7, 100 and 0.2, respectively. In terms of the silhouette settings, the offline pre-trained model (Qin et al., 2022) is used to provide the instance masks.

For the Gaussian Splatting, we use Adam for parameter optimization (the same hyperparameter values are used across all subjects). We set the learning rate to 5e-3 for the position and 1.7e-2 for the scaling of 3D Gaussians and keep the same learning rates as 3D Gaussian Splatting (Kerbl et al., 2023) for the rest of the parameters. We train for 20k iterations, and exponentially decay the learning rate for the splat positions until the final iteration, where it reaches $0.01 \times$ the initial value. We enable adaptive density control with binding inheritance every 2,000 iterations. The overall training time is around 6 hours on a single 4090Ti.

## 7.2 SPLITTING AND FUSION ALGORITHM

Algorithm 2 is the specific algorithm we use for superquadric splitting, can be visualized as Fig. 6. Additionally, Algorithm 3 is the specific algorithm we use for superquadric fusion.

## 7.3 FAILURE CASES

We also include some failure cases in Fig. 7(a). As discussed in the Limitations section, our method may encounter failure cases when fitting detailed structures, such as woods and leaves, due to the relatively coarse and mid-level nature of the superquadric representation. In future work, we aim to enhance our pipeline by augmenting superquadrics with a more diverse set of hybrid primitives, such as cuboids and curved surfaces, to better address these limitations.

---

**Algorithm 2** Split

---

**Input:** a set of clusters $\mathcal{S}_k$, a set of vertices $V_k$, a threshold for split $\tau$, max number of iterations $T$, a norm factor $\gamma$

1: **if** $N \geq 2$ **then**
2:      $D \leftarrow 0_{N \times N}$
3:      **for** $i = 1$ **to** $N - 1$ **do**
4:          $\mathcal{M}_{k,i} = \{m | A_{k,m} \in S_{k,i}\}$
5:          $\bar{A}_{k,i} \leftarrow \frac{1}{M} \sum_{m \in \mathcal{M}_{k,i}} (\Pr[\tilde{n}_{k,m} | A_{k,m}] \cdot A_{k,m})$            ▷ weighted average among $S_{k,i}$
6:          $m_{k,i} = \arg\min_{m \in \mathcal{M}_{k,i}} \mathbf{d}(A_{k,m}, \bar{A}_{k,i})$            ▷ find the centroid of $S_{k,i}$
7:          **for** $j = i + 1$ **to** $N$ **do**
8:              $\mathcal{M}_{k,j} = \{m | A_{k,m} \in S_{k,j}\}$
9:              $\bar{A}_{k,j} \leftarrow \frac{1}{M} \sum_{m \in \mathcal{M}_{k,j}} (\Pr[\tilde{n}_{k,m} | A_{k,m}] \cdot A_{k,m})$      ▷ weighted average among $S_{k,j}$
10:             $m_{k,j} = \arg\min_{m \in \mathcal{M}_{k,j}} \mathbf{d}(A_{k,m}, \bar{A}_{k,j})$        ▷ find the centroid of $S_{k,j}$
11:             $D_{i,j} \leftarrow \mathbf{d}(p_{k,m_{k,i}}, p_{k,m_{k,j}})$        ▷ Calculate the distance between $S_{k,i}$ and $S_{k,j}$
12:          **end for**
13:      **end for**
14:      $i^*, j^* \leftarrow \arg\max_{i,j} D_{i,j}$            ▷ find the two clusters with largest distance
15:      **if** $D_{i^*,j^*} > \tau$ **then**            ▷ split if distance is larger than the threshold
16:          $\alpha'_1 \leftarrow \alpha_k$ and $\alpha'_2 \leftarrow \alpha_k$
17:          $\alpha_k = 1$
18:          $\varepsilon'_1 \leftarrow \gamma \varepsilon_k$ and $\varepsilon'_2 \leftarrow \gamma \varepsilon_k$
19:          $r'_1 \leftarrow r_k$ and $r'_2 \leftarrow r_k$
20:          $s'_1 \leftarrow 0.4 s_k$ and $s'_2 \leftarrow 0.4 s_k$
21:          $v^*_{k,1} \leftarrow V_{k,m_{k,i^*}}$ and $v^*_{k,2} \leftarrow V_{k,m_{k,j^*}}$
22:          $t'_1 = v^*_{k,1} - [(s'_1 \cdot (\sum V_k)/|V_k|) \cdot \mathbf{rot}(r'_1)]$
23:          $t'_2 = v^*_{k,2} - [(s'_2 \cdot (\sum V_k)/|V_k|) \cdot \mathbf{rot}(r'_2)]$
24:          freeze the parameters of other blocks
25:      **end if**
26: **end if**

---

**Algorithm 3** Fuse

---

**Input:** all the vertices $\{V_k | k = 1, \ldots, K\}$, all the bounding boxes $\{B_k | k = 1, \ldots, K\}$, image embedding $h^{(\mathbf{I})}$

1: $k' \leftarrow \arg\min_{k' \in \{1,\ldots,K\}} \mathbf{d}\left((\sum_{v_k \in V_k} v_k)/|V_k|, (\sum_{v_{k'} \in V_{k'}} v_{k'})/|V_{k'}|\right)$      ▷ find the nearest neighbor
2: $\bar{B} = \max(B_k, B_{k'})$            ▷ merge bounding boxes
3: $\mathcal{A}_k = \varnothing$
4: **for** $p_i \in [P_k, P_{k'}]$ **do**
5:      $h_i^{(p)} \leftarrow f^{(p)}([p_i, \bar{B}])$            ▷ encode point and box prompt
6:      $A_i = \mathrm{CrossAttention}(h^{(\mathbf{I})}, h_i^{(p)})$            ▷ get cross attention map
7:      $\mathcal{A}_k \leftarrow \mathcal{A}_k \cup \{A_i\}$
8: **end for**
9: $\mathcal{S}_k \leftarrow \mathrm{HDBSCAN}(\mathcal{A}_k)$            ▷ cluster $\mathcal{A}_k$
10: **if** $|\mathcal{S}_k| = 1$ **then**            ▷ fuse if only one cluster found
11:      $\alpha'_k = 0.5\alpha_k + 0.5\alpha_{k'}$
12:      $\alpha_k = 1$ and $\alpha_{k'} = 1$
13:      $\varepsilon' = 0.5\varepsilon + 0.5\varepsilon_{k'}$
14:      $r' = 0.5 r_k + 0.5 r_{k'}$
15:      $s' = 0.5 s_k + 0.5 s_{k'}$
16:      $t' = 0.5 v^*_k + 0.5 v^*_{k'}$
17:      freeze the parameters of other blocks
18:      **for** $t = 1$ **to** $T$ **do**
19:          optimize the new fused block only
20:      **end for**
21: **end if**

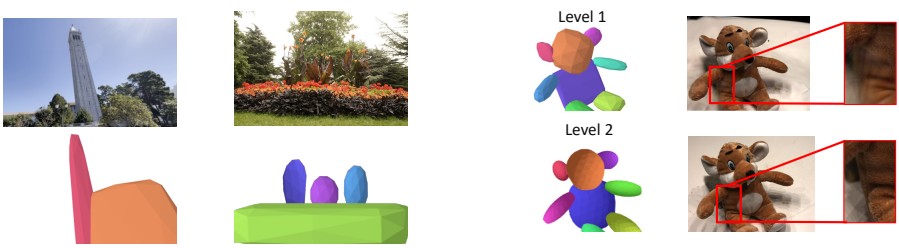

Figure 7: Failure case and Superquadric Resolution Comparison.

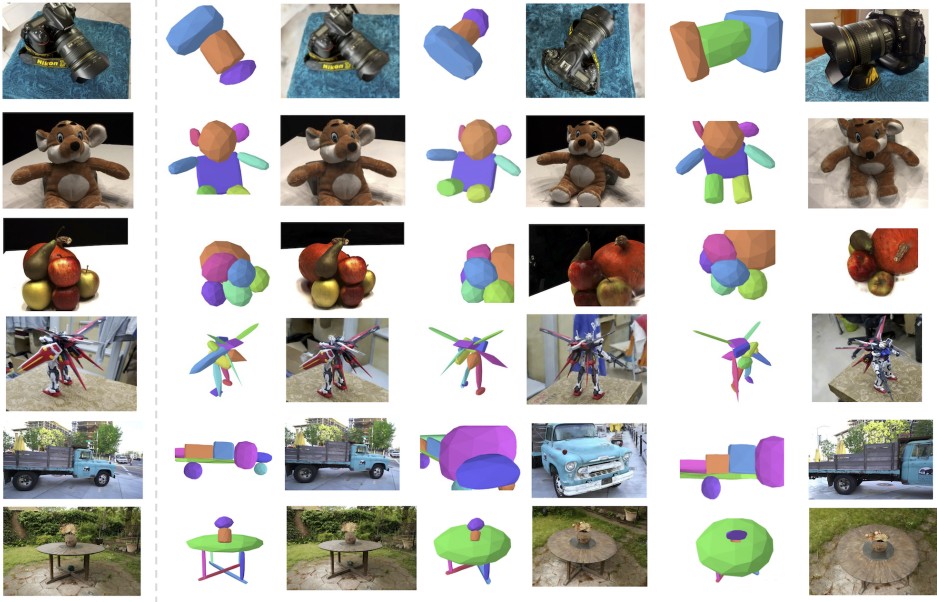

Figure 8: Multi-view reconstrucion results.
More multi-view reconstruction and editing results are shown in Fig. 8 and Fig. 9

## 7.4 SUPERQUADRIC RESOLUTION COMPARISON

As shown in Fig. 7, using a higher-resolution superquadric would enhance reconstruction quality. However, higher resolution also implies unfair comparisons with the baseline and highly increases the computational and memory overhead. In the future, we plan to further explore the integration of higher-resolution and more detailed primitives into our pipeline to achieve better results while maintaining efficiency.

## 7.5 MULTI-VIEW RESULTS

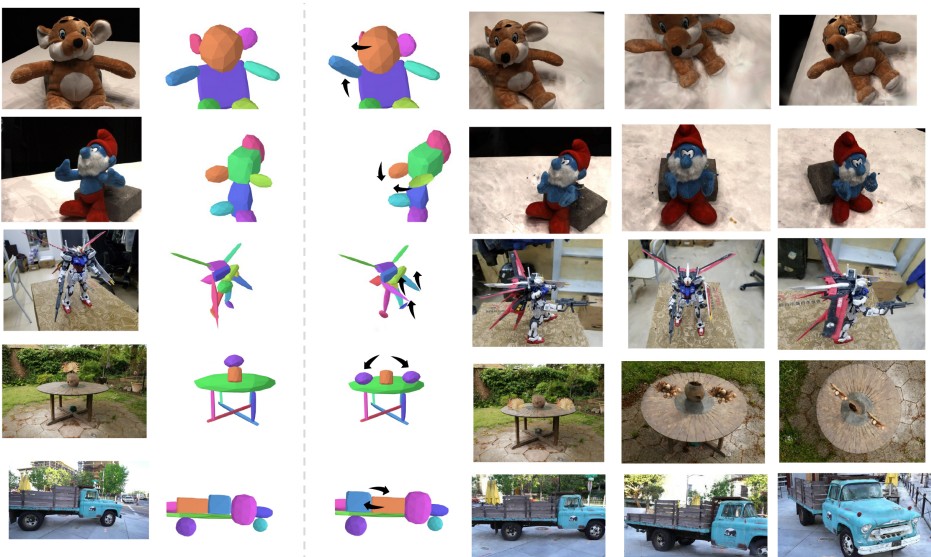

Figure 9: Multi-view editing results.

