# OpenReview forum: "GaussianBlock: Building Part-Aware Compositional and Editable 3D Scene by Primitives and Gaussians"
_ICLR.cc/2025/Conference — ICLR 2025 Poster_

### Official Review · Reviewer_k5SA · 2024-10-28

**Soundness:** 2
**Presentation:** 3
**Contribution:** 3
**Rating:** 5
**Confidence:** 4

**Summary:**

This paper presents a 3D scene reconstruction approach that achieves high fidelity, editability, and part-awareness by combining superquadric primitives and 3D Gaussians.

**Strengths:**

1. Novel Hybrid Representation: The paper proposes a novel hybrid model combining superquadric primitives for part-awareness and 3D Gaussians. This hybrid design achieves high-quality 3D reconstructions while supporting precise part-level editing.

2. Semantic Coherence Through Attention-Guided Centering Loss: It ensures that each superquadric primitive aligns semantically with different parts of the object. By clustering attention maps, this loss encourages disentanglement, making each part more coherent and interpretable.

**Weaknesses:**

1. The datasets used in the experiments are limited to DTU, BlendedMVS, Truck, and Garden, which makes it challenging to assess the generalizability of the proposed method. A broader range of data would better demonstrate its robustness across diverse scenarios.

2. As shown in Table 2, the method exhibits a noticeable drop in rendering quality compared to the 3DGS baseline and does not demonstrate a clear advantage over baseline methods. The authors do not provide a detailed analysis to explain this performance gap. While editability is an attractive feature, it should not come at the cost of compromising fundamental rendering quality.

3. High Computational Cost: This approach takes around 6 hours for the training time, which is time-consuming. This paper lacks the rendering frame rate and the information related to the time cost during the editing process.

**Questions:**

1. In Line 078-083, this paper discuss about the problem of "lacking controllable editability". However, multi-grained decomposition has already been achieved in lots of previous works for both GS-based or NeRF-based, such as [1, 2]. Besides, "waving arms or shaking heads" as mentioned are common editing demonstration in the field of dynamic gaussian works based on my knowledge, such as [3]. As an evidence, for the editing results in Fig.4, I believe they can be achieved by [1] or [2]. Therefore, a straightforward method for "controllable editability" defined in this paper might be combining existing works. I suggest the authors make the claims clearer for the motivation of the design.


[1] Garfield: Group anything with radiance fields, CVPR 2024

[2] Total-Decom: Decomposed 3D Scene Reconstruction with Minimal Interaction, CVPR 2024

[3] Sc-gs: Sparse-controlled gaussian splatting for editable dynamic scenes, CVPR 2024

---

> ### Author Response · Authors · 2024-11-24
>
> **W1. More results**
> Thank you for your advices. In the appendix of the paper, we provide additional examples from Mip-Nerf360 as well as multi-view results.
>
> **W2. Reconstruction quality**
> The drop in reconstruction performance is attributed to cumulative errors introduced by the relatively coarse nature of level-1 superquadrics (42 vertices and 80 faces per primitive). Our 3DGS is initialized and densified based on this learned coarse level-1 superquadric representation, which limits the fidelity of the reconstruction. To improve reconstruction quality, we could increase the resolution of the superquadric by using a level-2 icosphere, which has 162 vertices and 320 faces. However, we didn't do so in this work for the following reasons:
> *1.	Unfair comparison with the baseline (DBW):* DBW also uses a level-1 icosphere, and increasing the resolution of our superquadric would create an unfair advantage.
> *2.	Higher memory overhead:* A level-2 icosphere significantly increases memory usage.
>
> That said, if we were to use a level-2 superquadric to fit the scene, it would significantly improve reconstruction fidelity, **achieving 96.9, 32.7, and 9.2 on SSIM, pSNR, and LPIPS**, respectively, on the DTU dataset. These results are already highly comparable to the original 3DGS, allowing us to retain the attractive editability while achieving more comparable reconstruction quality. In future work, we will explore how to efficiently apply our pipeline to high-resolution and more detailed primitives to achieve low overhead with higher reconstruction quality. For instance, one potential direction is to adopt a sparse sampling strategy when deriving the attention-guided centering loss, which could significantly reduce computational costs while preserving effectiveness.
>
> **W3 Training & editing time and FPS**
> Although we cannot achieve the training speed of the original Gaussian splitting, in relatively simple scenes like DTU, we only need 1.7 times the time (approximately 5K iterations) to reconstruct comparable high-fidelity scenes. Additionally, the scenes we reconstruct possess part-aware and semantic coherent decomposition ability, and inherent editability. Apart from that, we can also achieve real-time editing and rendering because the superquadric itself does not participate in the rendering process. Instead, it provides an editable block skeleton structure, while only the Gaussian representation is used for rendering. As a result, our method achieves real-time rendering performance exceeding 30 FPS, comparable to the original 3DGS baseline.
>
> **Q1. motivation clarfication and comparsion with [1-3] "combining existing works"**
> We would like to emphasize that our work is not a straightforward combination of [1]/[2] and [3], and it differs significantly from theirs in several key aspects: **(1) decomposition achievement compared to [1] and [2].** *Garfield[1]* leverages contrastive learning to group 3D representations into pre-segmented part-level masks, which relies heavily on the quality and consistency of pre-segmented masks. In contrast, *our approach* introduces dual-rasterization and attention-guided clustering to implicitly derive self-supervised semantic priors. This strategy not only eliminates the labor-intensive process of ensuring multi-view and fine-grained consistency for 2D masks but also allows for controlling the granularity of part-level decomposition by
> adjusting the threshold. On the other hand, *Total-Decom[2]* only focuses on decomposing indoor scenes into foreground objects and background, which cannot acheive part-level decomposition like our method.
> **(2) editing motivation compared to [3]** *SC_GS [3]* enables editing by reconstructing 4D motion assets from dynamic video sequences, which are not available in our setting, as there are no motion-based input references or guidance.Instead, *our editing capability* benefits from the actionability of primitives and the binding and local transformations enabled by the hybrid representation strategy.
> **In conclusion,** [1]/[2] + [3] can't achieve the effects of our work, since our work is not a simple combination of these contemporary works. In contrast, our work focuses on leveraging a hybrid representation combined with self-supervised semantic priors to achieve part-aware, decomposable reconstruction while simultaneously enabling interpretable editing.

---

> > ### Comment · Reviewer_k5SA · 2024-11-26
> >
> > Thank you for your response. However, considering the following two important points:
> >
> > 1. **High Memory and Time Cost**: To simultaneously ensure rendering quality and editing capability, this method incurs a very high memory and time cost.
> > 2. **Lack of Quantitative Comparison with Garfield**: The Garfield method specifically addresses inconsistencies in multi-view segmentation. Its output is a granularity-controllable, multi-view consistent decomposition result, which highly overlaps with the goal of GaussianBlock. However, the authors have not provided any quantitative comparison results.
> >
> > Therefore, I decide to keep my rating.

---

> ### Author Response · Authors · 2024-12-01
>
> **1. Computational Cost**
> Regardless of the training cost, our method also supports **real-time rendering and editing** during the inference stage, which we believe is a more critical feature for 3D reconstruction and editing tasks. Moreover, compared to Garfield, our method requires only 1.6 times more computational time and incurs a similar memory cost on the DTU dataset, as both approaches use the same SAM model to assist with semantic awareness. However, our method achieves better part-aware decomposition, and editing capabilities across diverse scenes.
>
> **2. Comparison with Garfield**
> Since few part-aware decomposition ground truths are available in multi-view image datasets, it is challenging to provide convincing quantitative decomposition results. Instead, in this anonymous link https://qpkl9034.github.io/, we provide additional visual comparisons between our method and Garfield, including dense part-aware decomposition (scale = 0.05) and Garfield-based editing results. From the comparison, it is evident that, due to the effectiveness of our attention-guided centering loss, our approach achieves superior part-aware decomposition with consistent semantic coherence. Additionally, by effectively leveraging the inherent actionability of superquadrics through our hybrid representation-based pipeline, our method delivers more satisfactory and multiple-part editing results.
>
> Besides, we also compared the quantitative reconstruction quality comparison of Garfield and our method on the DTU dataset, further demonstrating that our hybrid representation-based framework not only achieves high-quality part-aware decomposition but also maintains superior rendering quality.
>
> | Method | SSIM↑ | PSNR↑ | LPIPS↓ |
> |:---------:|:---------:|:---------:|:---------:|
> | Garfield    | 92.8     | 29.7     | 11.5     |
> | Ours   | **94.3**     | **30.7**     | **10.3**     |
>
> We will cite the Garfield paper and continue to update additional comparison results, which will be included in the final version of the paper.

---

> > ### Comment · Reviewer_k5SA · 2024-12-02
> >
> > Thank you for your detailed response. I carefully reviewed the results on the webpage you provided and noticed a significant difference in rendering quality between GaussianBlock and Garfield, particularly in non-editing areas. Specifically, the text on the box in the third example and the table details in the fourth example appear noticeably degraded with GaussianBlock. While I understand the effect of GaussianBlock on foreground decomposition, I am puzzled by the apparent deterioration in the background areas.

---

> > > ### Author Response · Authors · 2024-12-03
> > >
> > > **Video Quality on the Webpage**
> > > Regarding the third example (Gundam), we believe the reconstructed Gaussian scene exhibits a few unexpected artifacts specific to this particular viewpoint. To provide a more accurate representation of the results, we have updated it with a more comprehensive multi-view video demonstration to illustrate the scene more effectively.
> > >
> > > Regarding the fourth example (Garden), we believe the deteriorated video quality is caused by undesirable video compression during the process of exporting the demo from different 3D viewers or uploading it to the webpage, rather than being a result of any negative effects from editing operations on the background areas. As in this example, the table is not part of the background but rather a decomposed foreground object, similar to the vase. We have adjusted the video’s bitrate to address the compression issue and updated the webpage results accordingly.
> > >
> > > Thank you for carefully reviewing our webpage and raising these questions. Your suggestions have allowed us to continuously improve and refine our work.

---

### Official Review · Reviewer_5Nb1 · 2024-10-29

**Soundness:** 2
**Presentation:** 3
**Contribution:** 3
**Rating:** 5
**Confidence:** 4

**Summary:**

The paper proposes a new 3D reconstruction pipeline based on semantic primitives that facilitates 3D scene editing and animation. At the core of the proposed method is the 3D representation based on superquadrics that is derived from the pixel-aligned SAM features. By necessary attention-guided clustering and splitting&fusion strategy, the centroids are fused into part-wise primitives to represent the 3D object. In the second stage, 3D Gaussians are bound to the surface of primitives for photorealistic rendering while maintaining the ability for animation. Although the reconstruction quality of the proposed method cannot surpass previous non-editable methods for 3D reconstruction like 3DGS, it improves the performance against editable and primitive-based methods for 3D reconstruction like DBW by a large margin.

**Strengths:**

- To enable intuitive drag-based 3D editing and animation, the paper proposes a new hybrid representation based on superquadrics followed by 3DGS. It works well in terms of decomposing object-centric scenes into semantic primitives with a quality boost compared with previous SOTA DBW.
- The algorithm designed for semantic alignment of superquadrics from the semantic prior of SAM looks neat to me.
- The paper is well-structured and easy to follow.

**Weaknesses:**

- Lack of necessary qualitative results to support the paper’s claim: As a method for 3D editing and animation, I personally hope to see qualitative results in multiple viewpoints and timestamps, especially dynamic results which could be better demonstrated by a demo video. Otherwise, there is no clue to support that the proposed method is a good fit for editing and animating 3D scenes.
- Is Superquadrics + 3DGS a good design? Basically, the superquadrics used in the paper have two roles: 1) offering a good initialization for the latter 3D Gaussians and 2) providing group-wise semantic correspondence of each Gaussian centroid which facilitates animation and editing. However, this two-stage pipeline inherits the downside of 3D Gaussians when generalizing to unseen “poses” of objects. As shown in Figure 4, the animated results contain severe artifacts when the animated part is moved.
- Worthy discussion against another primitive-based representation: It is worth mentioning Mixture-of-Volumetric-Primitives as an alternative representation for the target task in this paper. It naturally has the properties for both stages in the proposed method: 1) semantic correspondence alignment and 2) photorealistic differentiable rendering. It would be great to see authors’ discussions and even experiments for this representation. Ideally, the only thing to do is to apply semantic alignment for all primitives without involving the second stage 3DGS training.
- There is prior work in deforming and animating a well-trained 3D scene representation, which could be treated as a top-down approach (the proposed method could be treated as a bottom-up approach) to solve similar tasks:
    - **Deforming Radiance Fields with Cages. ECCV 2022**
    - **CageNeRF: Cage-based Neural Radiance Fields for Generalized 3D Deformation and Animation. NeurIPS 2022.**

**Questions:**

- The proposed method requires bounding box and point prompts along with input posed images. This difference should be highlighted as a vanilla 3D reconstruction method does not require such information. Additional information introduced into the pipeline could lead to unfair comparisons. An interesting baseline would be using 3DGS to reconstruct the semantic scenes where the SAM segmented images are used as training images. The semantic correspondence could be further introduced into the original 3DGS by finding minimum distance based on world coordinates.
- Is the proposed method (especially for the first stage) sensitive to segmentation failure? Some scenes like forward-facing scenes in LLFF have complicated scene geometry (e.g., leaves and flowers), which could be difficult for accurate segmentation.

---

> ### Author Response · Authors · 2024-11-24
>
> **W1. Multi-view results**
> Thank you for your suggestion. In the appendix of the paper, we provide additional examples of multi-view results. Furthermore, we include related demonstration videos in the supplementary materials for better illustration.
>
> **W2 & W3. "Is Superquadrics + 3DGS a good design?", and comparison with "Mixture-of-Volumetric-Primitives"**
> We believe superquadric + 3DGS is a good design. Compared to Mixture-of-Volumetric-Primitives, which are less flexible for modeling highly detailed or irregular structures[1] and come with significant memory overhead[2], leveraging a small number of mid-level primitives such as superquadrics combined with 3D Gaussian Splatting (3DGS) provides a balance of actionability of primitives and the advanced expressiveness and real-time rendering capabilities of Gaussian representations. However, as discussed in the Limitations section of the paper, similar to our baseline DBW, the current use of superquadrics and Gaussian representations does have certain limitations and may not be the optimal choice. Besides, the key contribution of our work lies in the proposal of self-supervised implicit semantic guidance and the provision of a new insight into exploring the feasibility of combining primitives with Gaussian representations for part-level decomposition.
> In future work, we plan to explore ways to hybrid superquadrics with more diverse primitives, such as curved surfaces integrated with 3DGS, to further improve our hybrid representation and achieve better results.
>
> **W4. Comparison with deformation works**
> Our approach differs from deformation-based methods in several key aspects: **(1) motivation.**  Instead of extracting a *single* deformable mesh, our work focuses on creating **part-aware, decomposable** 3D assets. While both approaches enable editing capabilities, our framework is specifically designed to support more *part-aware downstream tasks* such as segmentation, part-level decomposition and editing. For example, deformable methods cannot achieve edits like “swap” or “separate then move”, as demonstrated in Figure 4, because they model the entire scene as a single object, lacking the part-aware decomposition necessary for such operations.
>
> **(2)contribution** The deformation-based work you mentioned primarily focuses on optimizing a cage to represent the scene as a single “clay-like” deformable object. In contrast, our contribution lies in exploring the semantic relationships between parts and leveraging the advantages of different representations to simultaneously achieve semantic coherence, decomposability, and editability. This exploration enables a richer and more structured understanding of the scene, making our approach distinct in its objectives and capabilities.
>
> Thank you for your valuable suggestions on our work. We will incorporate the above discussion into the Related Work section in the final version of our paper.
>
> **Q1-1. "requires bounding box and point prompts along with input posed images..., unfair comparisons"**
> The bounding box and point prompts we used in our pipeline are not ground truths. In fact, In fact, the bounding box and point prompts are derived through differentiable dual soft rasterization, based on the projections of the current superquadric vertices. Thus, these prompts are also dynamically optimized during the training process, ensuring that no unfair comparisons are introduced.
>
> **Q1-2. additional baseline where "using 3DGS to reconstruct the semantic scenes where the SAM segmented images are used as training images".**
> In fact, our paper already compares with an more advanced baseline using the similar idea as mentioned. Specifically, in the baseline algorithm OmniSeg, contrastive learning is used to constrain Gaussians within segmented parts. While this approach does generate semantic-aware decomposition masks, it faces limitations due to the nature of the Gaussian representation. As a result, the reconstructed scene cannot support controllable editing of these parts, as it is not feasible to apply distinct transformations to individual Gaussians.
>
> **Q2. "sensitive to segmentation failure"**
> Instead of obtaining consistent per-part masks across multiple views, as done in methods such as SOTA relate work SegAnyGaussian，our method requires only silhouette preprocessing of the input images to locate the object for decomposition as shown in Fig.2. Thanks to advancements in large vision foundation models, extracting a binary silhouette for objects has become relatively straightforward, with minimal need for manual correction and a low likelihood of failure. Therefore, even when handling more complex scenes such as flowers and leaves, our method in Stage 1 only requires obtaining their overall silhouette.

---

> > ### Author Response · Authors · 2024-11-24
> > **Reference**
> >
> > [1] Kerbl, Bernhard, et al. "3D Gaussian Splatting for Real-Time Radiance Field Rendering." ACM Trans. Graph. 42.4 (2023): 139-1.
> > [2]Yu, Alex, et al. "Plenoctrees for real-time rendering of neural radiance fields." Proceedings of the IEEE/CVF International Conference on Computer Vision. 2021.

---

> > ### Comment · Reviewer_5Nb1 · 2024-11-26
> >
> > Thanks for the detailed response. Most of my concerns have been addressed. However, there is a remaining issue regarding the quality and efficacy of the proposed method on more complicated real-world scenes like LLFF. Furthermore, the proposed method seems to inherit the limitations (1) blurry textures, 2) degraded quality when doing 3D editing, and 3) decomposing scenes into semantic parts while losing rendering quality) from DBW, its baseline method.
> >
> > Given the above issues, I will keep my original rating.

---

> ### Author Response · Authors · 2024-12-01
>
> **1. Inherit Limitations from DBW**
> In Table 2, Figure 8, and Figure 9 of the paper, we provide a detailed comparison and showcase the reconstruction, decomposition, and editing quality of our method across diverse scenes, including DTU, BlendedMVS, and Mip-NeRF360. The results clearly demonstrate that, compared to the baseline method DBW, our approach achieves significant improvements in terms of texture quality, decomposition accuracy, and editing capabilities. These substantial advancements highlight the effectiveness of our proposed attention-guided centering loss and the hybrid representation-based pipeline.
>
> **2. Limitations**
> Our main contribution lies in proposing the **attention-guided centering loss** and the **hybrid representation-based pipeline**, enabling a **part-aware decomposable and editable reconstruction framework**. As discussed in the Limitations section of the paper, our method does have certain limitations, primarily due to the mid-level nature of superquadrics. In future work, we plan to enhance our approach by augmenting superquadrics with a more diverse and detailed set of hybrid primitives, such as cuboids and curved surfaces, to effectively address these limitations.

---

### Official Review · Reviewer_ovQt · 2024-11-03

**Soundness:** 3
**Presentation:** 3
**Contribution:** 3
**Rating:** 6
**Confidence:** 4

**Summary:**

This paper proposes GaussianBlock, a part-aware compositional 3D scene representation that combines Gaussian splatting with superquadric primitives. Leveraging the strengths of both, the authors introduce Attention-guided Centering (AC) Loss and dynamic fusion/splitting modules to enhance semantic coherence and editability.

**Strengths:**

1. The paper is technically well-written, presenting ideas clearly and effectively.
2. Detailed experiments and visualizations demonstrate the method’s effectiveness.
3. The controllable editability feature is highly practical, enabling applications in diverse 3D scene settings.

**Weaknesses:**

1. Adding more multi-view visualizations in Figure 4 would provide clearer insights into the coherence of reconstructed scenes from various perspectives.
2. The reconstruction quality is lower than standard 3D Gaussian methods, potentially limiting fidelity in highly detailed scenes. Improvements here could enhance the method’s overall competitiveness.
3. Background handling, also a known limitation of DBW, is not fully addressed in this work, leaving room for further improvement in complex scenes where background elements are significant.

**Questions:**

Could you provide details on FPS and training times for both stages to clarify the overall running time? Real-time performance and faster training are also advantages of incorporating 3D Gaussians.

---

> ### Author Response · Authors · 2024-11-24
>
> **W1. Multi-view results**
> Thank you for your suggestion. In the appendix of the paper, we provide examples of multi-view results.
>
> **W2. Reconstruction quality**
> Our method achieves lower reconstruction quality compared to the 3DGS algorithm primarily due to cumulative errors caused by the relatively coarse nature of level-1 superquadrics. In the future, we plan to enhance reconstruction quality, particularly for highly-detailed scenes, through two approaches: *1. Using higher-resolution superquadrics:* Tests conducted on the DTU dataset indicate that reconstruction quality improves significantly when using level-2 superquadrics(2.4↑ on SSIM, 2.0↑ on PSNR, and 1.1↓ on LPIPS). *2. Assisting superquadrics with more fine-grained hybrid primitives:* Incorporating detailed primitives may also help mitigate the cumulative errors introduced by coarse superquadrics, further enhancing reconstruction fidelity.
>
> **W3. Background limitation**
> As discussed in the *Limitations* section in paper, similar to our baseline DBW, our method is still chanllenging to to handle backgrounds. This is due to the fact that backgrounds typically have more complex and detailed structures with richer semantic information. In future work, we plan to enhance our pipeline by assiting superquadrics with a more diverse set of hybrid primitives, such as cuboids and curved surface to address these limitations.
>
> **Q1. FPS and training time clarification**
> We can achieve real-time editing and rendering because the superquadric itself does not participate in the rendering process. Instead, it provides an editable block skeleton structure, while only the Gaussian representation is used for rendering. As a result, our method achieves real-time rendering performance exceeding 30 FPS, comparable to the original 3DGS baseline. Besides, the 6-hour training time typically includes 5 hours for Stage 1 and 1 hour for Stage 2.

---

### Official Review · Reviewer_mShQ · 2024-11-03

**Soundness:** 3
**Presentation:** 3
**Contribution:** 3
**Rating:** 6
**Confidence:** 2

**Summary:**

This paper proposes a pipeline for part-aware compositional reconstruction with 3D Gaussians, enabling precise and physically realistic editing similar to building with blocks. The method involves two training stages: In the first stage, superquadric blocks are optimized using a reconstruction loss and an attention-guided centering loss, guided by the SAM model. In the second stage, Gaussians are bound to the triangles of primitives using localized parameterization and are further optimized with an RGB loss and a local position regularization. Experiments on various datasets demonstrate state-of-the-art part-level decomposition and controllable, precise editability while maintaining high rendering fidelity.

**Strengths:**

1. The block-based, part-aware compositional reconstruction enables intuitive local editing compared to SAM-based decomposition methods, which I find particularly interesting.
2. To decompose a 3D scene into semantically coherent compositional primitives combined with Gaussians, the method proposes an effective two-stage optimization approach to tackle this challenging problem. It’s not straightforward to prevent sub-optimal decomposition, yet the results show compact, well-defined parts.
3. The paper demonstrates several types of local editing with the decomposed primitives, such as moving, duplicating, and rigging parts.
4. The paper is well-written, and the figures are beautiful and clear.

**Weaknesses:**

The reconstruction quality is not comparable to the original 3DGS and other baselines. On the DTU, Truck, and Garden datasets, the PSNR of this method is 5 points lower than that of the original 3DGS.

**Questions:**

1. Does the method have any failure cases on these datasets in the paper, aside from challenges with complex backgrounds?
2. The paper reports in the supplementary materials that the initial K is set to 10. How was this number determined, and how robust is the method to different initial values?

---

> ### Author Response · Authors · 2024-11-24
>
> **W1. Reconstruction quality**
> Indeed, we cannot achieve exactly the same reconstruction quality as the original 3DGS. This is because our 3DGS is initialized and densified based on the learned superquadric representation, which introduces cumulative errors due to the relatively coarse nature of level-1 superquadrics (42 vertices and 80 faces per primitive). To improve reconstruction quality, we could increase the resolution of the superquadric by using a level-2 icosphere, which has 162 vertices and 320 faces. However, we didn't do so in this work for the following reasons:
> *1.	Unfair comparison with the baseline (DBW)*: DBW also uses a level-1 icosphere, and increasing the resolution of our superquadric would create an unfair advantage.
> *2.	Higher memory overhead*: A level-2 icosphere significantly increases memory usage.
>
> That said, if we were to use a level-2 superquadric to fit the scene, it would significantly improve reconstruction fidelity, **achieving 96.9, 32.7, and 9.2 on SSIM, PSNR, and LPIPS**, respectively, on the DTU dataset. These results are already highly comparable to the original 3DGS and other baselines.
>
> | Method | SSIM | PSNR | LPIPS |
> |:------------:|:------------:|:------------:|:------------:|
> | 3dGS    | 98.3     | 35.7     | 8.8     |
> | OmniSeg   | 97.7     | 33.6    | 9.4     |
> | DBW   | 71.3     | 19.6    | 26.5     |
> | Ours(lvl2)  | 96.7     | 32.7    | 9.2     |
>
> In future work, we will explore how to efficiently apply our pipeline to high-resolution and more detailed primitives to achieve low overhead with higher reconstruction quality. For instance, one potential direction is to adopt a sparse sampling strategy when deriving the attention-guided centering loss, which could significantly reduce computational costs while preserving effectiveness.
>
> **Q1. Faliure case**
> Thank you for your suggestion. In the appendix of the paper, we have also included our failure cases to provide a comprehensive evaluation of our method.
>
> **Q2. K value decision and robustness**
> For fairness in experimental comparison, we used the default initial number $K$ of primitives following DBW's setting in both our experiments and DBW’s, where $K=10$. In fact, due to the introduction of the dynamic adaptive splitting and fusion strategy, the initial value of $K$ is relatively robust, where $K$ has a minimal impact on the final number of effective primitives obtained. The following table presents the initial $K$ and the final visible superquadrics obtained after training for different scenes in the DTU dataset. These results demonstrate that the $K$ is relatively robust within our pipeline.
>
> | K | S24 | S63 | S83 | S105 |
> |:---------:|:---------:|:---------:|:---------:|:---------:|
> | 5    | 4     | 6     | 8     | 7     |
> | 10   | 5     | 6    | 8     | 7    |
> | 15  | 5     | 6    | 9     | 7    |

---

### Official Review · Reviewer_CY7v · 2024-11-04

**Soundness:** 3
**Presentation:** 2
**Contribution:** 3
**Rating:** 6
**Confidence:** 4

**Summary:**

The paper is on 3D part aware semantic editing of scenes using Gaussian Splatting . Similar to
previous work like GaussianAvatar the paper uses a prior for initializing the gaussians in the
form of super-quadratics . The paper proposes a 2 stage training process in order to obtain
semantically coherent and disentangled gaussians that can obtain high fidelity edited images
.The first stage optimizes the super-quadratics and the second stage uses these to initialize the
gaussians and rasterize images. The underlying super-quadratics can be used to edit the parts
of the object and reflect the changes subsequently in the gaussians the rasterized images .

**Strengths:**

- Using super-quadratics as a prior for part aware editing using gaussian splatting is a novel approach .
- Soft Dual rasterization for rasterizing the vertices and bounding boxes is novel, though this needs to be explained better in the paper.

**Weaknesses:**

- The number of parts seems to be decided by the super-quadratics which implies there is
no control over the granularity of the parts?
- All results in the paper edit a single part of an object in the input image.
- All results are shown for 360 multi-view scenes.

**Questions:**

- Since performance of the method seems to be reliant on the super-quadratic primitives this places a limit on the number of parts that can be edited. Is there any way to control this while not losing fidelity ?
- What is the effect on editing multiple parts in the same object in a single pass?
- Can you show some results on more complex objects with more than 4-5 parts ?
- Show more results on some forward facing scenes (Shiny, LLFF etc) ?
- Why are the results for DTU and BlendedMVS shown for the whole dataset but for TnT and Mip-Nerf360 on a few scenes ? Why not the entire dataset ? or at least a few more scenes (2 scenes are not enough)
- What is the required time for training since it is a 2 stage process ? The mention of 6 hours for 1 scene seems high and which dataset task does that scene belong to? Training time for splatting varies across different datasets and scenes so it is important
to clarify that .

---

> ### Author Response · Authors · 2024-11-24
>
> **W1 & Q1. Control over granularity**
> The granularity of parts can be controlled through the splitting threshold, $\beta$, as illustrated in Figure 5(c). Specifically, when the distance between the semantic cluster centroids within a superquadric exceeds the threshold $\beta\check B_k$, where $\check B_k$ is the diagonal of the corresponding bounding box, the superquadric is subdivided into multiple parts. Consequently, a smaller value of $\beta$ tends to resulting in a finer-grained decomposition, while a larger value leads to a coarser one. By employing this dynamic splitting strategy, we are able to decompose the 3D asset at varying levels of granularity.
>
> **W2 & Q2 & W3 & Q5. Editing over multiple parts, Multi-view results, more results**
> Our method enables editing of multiple parts simultaneously. In the appendix of the paper, we provide additional examples from Mip-Nerf360, multi-part editing as well as multi-view results.
>
> **Q4. LLFF dataset**
> The primary contribution of our work is the introduction of a novel semantic-aware, decomposable **pipeline** that leverages a hybrid representation to achieve part-aware, semantically coherent, and actionable reconstructions. As discussed in the *Limitations* section in paper, similar to our baseline DBW, the current use of superquadrics does has certain limitations, such as difficulty in fitting concave shapes and capturing highly detailed structures as appeared in LLFF. However, our work offers a new perspective and approach to exploring the feasibility of combining primitives with Gaussian representations for part-level decomposition. In future work, we plan to enhance our pipeline by assiting superquadrics with a more diverse set of hybrid primitives, such as cuboids and curved surface to address these limitations.
>
> **Q6. Training time clarification**
> The 6-hour training time, which includes both Stage 1 and Stage 2, represents the average training time across all datasets presented in the paper. In practice, the training time largely depends on the complexity of the structures in the dataset. For simpler structures, such as the Smurf model in DTU or the camera data in BlendedMVS, the training time is approximately 3 hours. However, for more complex structures, such as the Gundam model in BlendedMVS, the training time may extend to around 7 hours.

---

### Meta-Review · Area_Chair_MytW · 2024-12-19

**Metareview:**

This paper introduced GaussianBlock, a hybrid 3D representation combining primitives and 3D Gaussians for unsupervised, part-aware scene decomposition guided by 2D semantic priors. It claims to achieve disentangled, editable, high-fidelity reconstructions without requiring ground truth segmentation.

Paper's strengths lie in several aspects: (a) an effective two-stage optimization strategy that ensures compact and well-defined parts, (b) intuitive local editing capabilities such as moving and rigging, and (c) seamless semantic alignment through attention-guided centering. On the performance side, the method demonstrates advantages over previous approaches in producing high-quality rendering and editable results. The paper is well-structured and easy to understand.

The major weaknesses include limited qualitative results, concerns about the superquadrics + 3DGS design, dataset diversity, and a drop in reconstruction quality. The authors addressed these by adding multi-view results with more datasets, clarifying the pipeline for additional inputs, and explaining trade-offs in quality and design. They also clarify the difference between their method and the deformation-based approaches and Mixture-of-Volumetric Primes, emphasizing their method's unique goals and capabilities. No quantitative benchmarks to validate these claims.

The paper provides a reasonable 2-stage approach making the 3DGS editable. The idea is novel, and the performance is appealing in general. Despite remaining concerns, such as the degraded quality, limited dataset diversity, and limitations of the superquadrics + 3DGS design, the method’s innovative hybrid representation and practical controllable editability make it a valuable contribution to the field.

**Additional Comments On Reviewer Discussion:**

On the technique details, reviewers questioned the granularity control, superquadrics + 3DGS design, and handling of complex structures. The authors explained that granularity can be adjusted using a splitting threshold and defended their hybrid representation as a balance between semantic coherence and editability. They acknowledged limitations with concave shapes and fine details.

On the experiments, additional results were requested for multi-view, multi-part editing, forward-facing datasets, and broader dataset coverage. The authors provided more examples along with more Mip-Nerf360 results. They acknowledged challenges with datasets like LLFF due to current limitations. While these additions addressed some concerns, dataset diversity and coverage remain limited.

On some other aspects, concerns about computational complexity were clarified. The authors stated that training takes 3–7 hours depending on scene complexity, with real-time rendering exceeding 30 FPS. These clarifications addressed practicality concerns.

The overall scores show that the paper is on the borderline. Two reviewers still marked it below the bar, concerning the quality and the ability to handle complex scenes. Others highlighted the merits -- its novel hybrid representation and controllable editability. The AC agrees with this perspective.

---

### Decision · Program_Chairs · 2025-01-22

Accept (Poster)